# Enhancing the Rice Seedlings Growth Promotion Abilities of *Azoarcus* sp. CIB by Heterologous Expression of ACC Deaminase to Improve Performance of Plants Exposed to Cadmium Stress

**DOI:** 10.3390/microorganisms8091453

**Published:** 2020-09-22

**Authors:** Helga Fernández-Llamosas, Juan Ibero, Sofie Thijs, Valeria Imperato, Jaco Vangronsveld, Eduardo Díaz, Manuel Carmona

**Affiliations:** 1Microbial and Plant Biotechnology Department, Centro de Investigaciones Biológicas-CSIC, Ramiro de Maeztu 9, 28040 Madrid, Spain; hfllamosas@gmail.com (H.F.-L.); juanibecab@gmail.com (J.I.); ediaz@cib.csic.es (E.D.); 2Department of Environmental Biology, Centre for Environmental Sciences, Hasselt University, 3590 Diepenbeek, Belgium; sofie.thijs@uhasselt.be (S.T.); valeria.imperato@uhasselt.be (V.I.); jaco.vangronsveld@uhasselt.be (J.V.); 3Department of Plant Physiology, Faculty of Biology and Biotechnology, Maria Skłodowska-Curie University, 20-0033 Lublin, Poland

**Keywords:** phytoremediation, endophyte, pollutant, growth-promotion, heavy metals

## Abstract

Environmental pollutants can generate stress in plants causing increased ethylene production that leads to the inhibition of plant growth. Ethylene production by the stressed plant may be lowered by Plant Growth-Promoting Bacteria (PGPB) that metabolizes the immediate precursor of ethylene 1-aminocyclopropane-1-carboxylate (ACC). Thus, engineering PGPB with ACC deaminase activity can be a promising alternative to mitigate the harmful effects of pollutants and thus enhance plant production. Here we show that the aromatics-degrading and metal-resistant *Azoarcus* sp. CIB behaves as a PGP-bacterium when colonizing rice as an endophyte, showing a 30% increment in plant weight compared to non-inoculated plants. The cloning and expression of an *acdS* gene led to a recombinant strain, *Azoarcus* sp. CIB (pSEVA237acdS), possessing significant ACC deaminase activity (6716 nmol mg^−1^ h^−1^), constituting the first PGPB of the Rhodocyclaceae family equipped with this PGP trait. The recombinant CIB strain acquired the ability to protect inoculated rice plants from the stress induced by cadmium (Cd) exposure and to increase the Cd concentration in rice seedlings. The observed decrease of the levels of reactive oxygen species levels in rice roots confirms such a protective effect. The broad-host-range pSEVA237acdS plasmid paves the way to engineer PGPB with ACC deaminase activity to improve the growth of plants that might face stress conditions.

## 1. Introduction

Ethylene is a gaseous plant hormone with essential functions in plant growth and development such as seed germination, cell expansion and elongation of stems and roots, the formation of adventitious roots and root hairs, fruit ripening, photomorphogenesis, and senescence in general [1,2,3,4,5,6]. However, the increase of ethylene concentrations in tissues as a response to environmental stress may cause adverse effects on plant health. A peak of ethylene activates defense systems in plant cells [7,8,9], but in case the stress is maintained, higher amounts of ethylene are produced, which can cause plant growth inhibition and even apoptosis or cell death [10,11]. Therefore, for normal growth and development of plants it is mandatory to keep the production of ethylene under control [6]. In vascular plants, ethylene biosynthesis is under the control of ACC synthases that convert *S*-adenosyl-l-methionine (SAM) to 1-aminocyclopropane-1-carboxylate (ACC) and 5′-methylthioadenosine (MTA), which is recycled to l-methionine. ACC is converted to ethylene by the action of ACC oxidases [6]. Increased ethylene production has been linked to a good number of biotic stresses such as pathogenic bacteria and fungi and insects. In addition, abiotic stresses such as drought, humidity, radiation, extreme temperatures, the presence of organic or inorganic pollutants such as aromatic compounds or metals and mechanical wounding can increase the levels of ethylene [7,11]. For instance, moderate concentrations of cadmium (Cd) induce oxidative stress, and consequently, ethylene production negatively affects plant growth-producing physiological, morphological, and biochemical disorders [12,13,14,15], limiting crop production and promoting agricultural losses [16].

The stress-induced increase of ethylene production in plants may be attenuated by some PGPB that are able to produce the ACC deaminase enzyme [6,17]. The ACC produced by plant roots is transported into the bacterial cytoplasm inducing the synthesis of ACC deaminase that cleaves ACC into ammonia and α-ketobutyrate that can be used by the bacteria as nitrogen and carbon source, respectively. As a result, the microbial metabolism reduces the ACC levels in the host plant, thereby diminishing the ethylene levels in the tissues and thus, the adverse effects on the plant [17,18]. The heterologous expression of ACC deaminase genes and their biological effects have been reported previously in some bacteria [19,20,21], although as far as we know the expression of an *acdS* gene in an endophytic bacterium to mitigate the negative effects of cadmium has not been shown.

Among PGPB, some β-Proteobacteria of the Azoarcus genus have been described. The genomes of three Azoarcus strains able to colonize plants as endophytes have been sequenced: *Azoarcus* sp. BH72 [22], *A. olearius* DQS-4 [23] and *A. communis* SWub3 (Acc. No. PRJNA234909). *Azoarcus* sp. BH72 and *A. communis* SWub3 were isolated from the roots of *Leptochloa fusca* [24], but the BH72 strain was also able to colonize rice and sorghum [25,26,27] and to induce a moderate promotion of plant growth [25,28,29]. *A. olearius* DQS-4 is closely related to strain BH72, which is capable of colonizing roots and aerial tissues of rice and green foxtail (*Setaria viridis*) and to promote their growth [23]. In addition to these three *Azoarcus* strains, we have characterized a related bacterium, *Azoarcus* sp. CIB, that is a facultative anaerobic (denitrifying) strain able to colonize rice roots as an endophyte, showing plant growth-promoting (PGP) traits [30], but its ability to promote rice growth has not been experimentally demonstrated. However, and in contrast to the typical *Azoarcus* strains, strain CIB is closely related to the new genus *Aromatoleum* [31] and it has some other interesting biotechnological properties such as the ability to degrade toxic aromatic compounds, e.g., toluene/xylene, under aerobic and anaerobic conditions [32,33,34,35], and to tolerate high concentrations of certain metals and metalloids [36] as well as to convert them into metal nanoparticles of high industrial value [37].

Heavy metals and metalloids are frequently found as pollutants in soils via different anthropogenic activities, e.g., smelting, mining, disposing of hazardous materials, and fertilization [38]. Cadmium is one of the most hazardous metals by the United States Environmental Protection Agency (US-EPA) [39] and, even at low concentrations in soils, can produce toxic plant effects, decreasing agricultural yields [40]. Some reports showed that rice can be a promising candidate for effective Cd phytoextraction in paddy soils [41,42,43]. The combination of plant-interaction abilities with those for biodegradation/tolerance to toxic compounds that might be present in soils makes *Azoarcus* sp. CIB a promising bacterial endophyte that could be engineered to promote rice growth under certain environmental stress conditions.

The aim of the present study was to investigate if *Azoarcus* sp. CIB is a PGP-bacterium and to increase its putative PGP abilities by the expression of an exogenous ACC deaminase, investigating the capability of the engineered CIB strain to enhance the growth of rice seedlings cultivated in Cd-polluted soils.

## 2. Materials and Methods

### 2.1. Strains, Seeds and Plasmids Used

The bacterial strains and plasmids used in this work are listed in Table 1. *Azoarcus* sp. strain CIB (NCBI taxonomy ID: 198107) was deposited in the Spanish Type Culture Collection (CECT #5669). Seeds from *Oryza sativa* L. GLEVA were kindly provided by Castells Seeds Co. (Tarragona, Spain).

### 2.2. Culture Media and Growth Conditions

*Azoarcus* sp. CIB was grown aerobically on MC medium (MA basal medium plus trace elements and vitamins) [32] at 30 °C. The composition of MA basal medium was the following, per liter of distilled water: 0.33 g of KH_2_PO_4_, 1.2 g of Na_2_HPO_4_, 0.11 g of NH_4_Cl, 0.1 g MgSO_4_ × 7H_2_O, 0.04 g of CaCl_2_ (pH 7.5) supplemented with trace elements [stock solution 100×; 1.5 g of nitrilotriacetic acid, 3 g of MgSO_4_ × 7H_2_O, 0.5 g of MnSO_4_ × 2H_2_O, 1 g of NaCl, 0.1 g of FeSO_4_ × 7H_2_O, 0.18 g of CoSO4 × 7H_2_O, 0.1 g of CaCl_2_ × 2H_2_O, 0.18 g of ZnSO_4_ × 7H_2_O, 0.01 g of CuSO_4_ × 5H_2_O, 0.02 g of KAl(SO_4_)_2_ × 12H_2_O, 0.01 g of H_3_BO_3_, 0.01 g of Na_2_MoO × 2H_2_O, 0.025 g of NiCl_2_ × 6H_2_O, and 0.3 mg of Na_2_ScO_3_ × 5H_2_O (pH 6.5) per liter of deionized water; vitamin solution (stock 1000×: 20 mg of biotin, 20 mg of folic acid, 10 mg of pyridoxine-HCl, 50 mg of thiamine-HCl × 2H_2_O, 50 mg of riboflavin, 50 mg of nicotinic acid, 50 mg of calcium d-pantothenic acid, 50 mg of vitamin B12, and 50 mg of *p*-aminobenzoic acid per liter of distilled water) [32]. As carbon source, 0.2% (*w/v*) pyruvate was added. When necessary, bacterial strains were also grown on a variant of the VM-ethanol rich medium [28] with the following composition (in 1 L of distilled water): KH_2_PO_4_ 0.4 g; K_2_HPO_4_ 0.6 g; NaCl 1.1 g; NH_4_Cl 0.5 g; MgSO_4_ × 7H_2_O 0.2 g; CaCl_2_ 26 mg; MnSO_4_ 10 mg; Na_2_MoO_4_ 2 mg; Fe(III)-EDTA 66 mg; yeast extract 1 g; bactopeptone 3 g; ethanol 6 mL (pH 6.8).

*E. coli* was grown at 37 °C in lysogeny broth (LB) medium [49] or in VM-ethanol medium [28] at 30 °C. When appropriate, kanamycin (50 µg mL^−1^) or gentamycin (10 µg mL^−1^) was added to the medium.

To determine the ACC deaminase activity a slightly reformulated DF medium [50] was used (per 1 L of distilled H_2_O): 4.0 g of KH_2_PO_4_; 6.0 g of Na_2_HPO_4_; 0.2 g MgSO_4_ × 7H_2_O; 2.0 g glucose and 2.0 g of pyruvate. Trace elements and vitamins were added at the same concentrations than used to formulate the MC medium. 3 mM 1-aminocyclopropane-1-carboxylic acid (ACC) was added as the sole nitrogen source.

### 2.3. Molecular Biology Techniques

Standard molecular biology techniques were performed as previously described [51]. All DNA fragments were purified using Gene-Turbo (BIO101 Systems, Irvine, CA, USA). Plasmids and PCR products were purified utilizing a High Pure Plasmid and PCR Product Purifications kits (Roche), respectively. Oligonucleotides were supplied by Sigma Co, and their sequences were: 5′RBSacdS (5′-CCAAGCTTTGACCTAAGGAGGTAAATAATGAACCTGCAACGATTCCCTCGTTAC-3′ (HindIII site underlined)); 3′acdS (5′-GGACTAGTTTAGCCGTTGCGGAAAATGAAGCTG-3′ (SpeI site underlined)); 5′PtacacdS (5′-TTGGCGCGCCCTGGCAAATATTCTGAAATGAGCTG-3′ (AscI site underlined)) and 3′PtacacdS (5′-TTGGCGCGCCTTAGCCGTTGCGGAAAATGAAG-3′ (AscI site underlined)). The cloned inserts and the DNA fragments employed were confirmed by DNA sequencing with fluorescently labeled dideoxynucleotide terminators [52] and AmpliTaq FS DNA polymerase (Applied Biosystems, Foster City, CA, USA) in an ABI Prism 377 automated DNA sequencer (Applied Biosystems). Transformations of *E. coli* were performed by using the RbCl method or by electroporation (Gene Pulser, Bio-Rad, Hercules, CA, USA) [51]. Transformation of *Azoarcus* sp. CIB was done by biparental conjugation using the strain *E. coli* S17-1λ*pir* as a donor, following a protocol previously established [44] with the following small modifications: donor cells were grown to *A_600_* of 5, and the receptor CIB strain was grown on MC medium supplemented with pyruvate 0.2%, and concentrated to reach an *A_600_* of 35. The transconjugants were selected on MC supplemented with 10 mM glutarate plus the corresponding antibiotic.

### 2.4. Construction of Plasmids pIZacds and pSEVA237acdS

The *acdS* gene was PCR-amplified by using the oligonucleotides 5′RBSacdS/3′acdS and the genomic DNA template from *P. phytofirmans* PsJN. The amplified fragment of 1017 bp was first cloned into plasmid pIZ1016 using the HindIII and SpeI restriction sites, rendering the plasmid pIZacdS (Table 1). A 1171-bp *Ptac-acdS* DNA fragment was PCR-amplified from pIZacdS plasmid using the oligonucleotides 5′PtacacdS/3′PtacacdS. The amplified fragment was cloned into wide-host-range plasmid pSEVA237 using the AscI restriction site, obtaining the plasmid pSEVA237acdS (Table 1).

### 2.5. Inoculation of Rice Seedlings with Bacteria

Dehulled rice seeds (*O. sativa* L. GLEVA) were surface sterilized by shaking for 30 min in 30 mL 1% (*v/v*) sodium hypochlorite. After rinsing them three times for 10 min in sterile water, the seeds were incubated in VM-ethanol for 48 h. The germ-free seedlings were selected for inoculation. The germination of the seeds was continued on humidified filter papers for 24 h prior to inoculation with *gfp*-expressing bacteria cells. The *gfp*-expressing *Azoarcus* sp. CIB (pSEVA237) or *E. coli* CC118 (pSEVA237) bacteria were obtained as described previously [30]. After growing up to the mid-exponential phase, bacterial cells were collected by centrifugation, washed with sterile 0.9% NaCl (*w/v*) solution, resuspended in 1 mL of sterile distilled water, and the cell suspension was inoculated onto the surface of each seedling in aseptic conditions. After inoculation, the seedlings were grown at 25 °C under natural daylight conditions (10 h of light and 14 h of darkness) in sterile tubes (12 mL volume and one seedling each tube) for 5–10 days.

### 2.6. Examination of the Growth of Inoculated Rice Plants in Climate Chambers

Dehulled surface sterilized rice seeds (*O. sativa* L. GLEVA) were obtained by shaking them at 25 °C for 30 min in 30 mL 1% (*v/v*) sodium hypochlorite. Seeds were inoculated with the corresponding strain, as described before. The inoculated seeds were sown in square pots (2 seeds per pot) of 72 × 72 × 100 mm (Kisker Biotech, Steinfurt, Germany) with autoclaved vermiculite (Soprema Iberia S.R.U., Madrid, Spain; grain size of size 0.5–3 mm) as solid substrate. The bacteria (*A*_600_ of 0.6) were additionally inoculated into the liquid Hoagland’s No.2 solution (Sigma-Aldrich, San Louis, MS, USA) used to water plants. As a control, non-inoculated seeds were germinated and cultivated in the same conditions. The plants were kept in the growth chamber under the following conditions: photoperiod of 18 h light (day) and 8 h of darkness (night); the temperature of 25 °C during the day and 22 °C during the night; humidity of 75% during the day and 80% during the night. At harvest time, the fresh weight of each plant was determined. Each experiment was performed by triplicate and the total number of plants used was 20.

### 2.7. Examination of the Growth of Inoculated Rice Plants under Greenhouse Conditions

Plants were inoculated with bacteria as described above. Twenty-four hours from the inoculation, the seedlings were introduced in test tubes (12 mL of volume) with semisolid (0.4% agar) Hoagland’s No.2 basal salt mixture (Sigma-Aldrich) supplemented, when required, with 50 µM CdCl_2_. Plants were watered every 5 days with sterilized Hoagland’s No.2 basal salt mixture. When required, 50 µM CdCl_2_ and bacterial suspensions (*A*_600_ = 0.6) were added to the irrigation solution. The growing plants were kept in a greenhouse at 30 °C for 15 days. After 15 days, root and shoot samples of 50–100 mg each were taken for metal extractions. The determined dry weights of these samples were used to calculate average root and leaf Cd concentrations per g dry weight. Each experiment was performed in triplicate and the number of plants used was 5.

### 2.8. ACC Deaminase Enzymatic Assay

The ACC deaminase activity was measured spectrophotometrically following a protocol based on the accumulation of α-ketobutyrate [53], with slight modifications. The bacterial culture was grown until the mid-exponential phase, the cells were gathered by centrifugation and the pellet was resuspended in DF medium supplemented with 3 mM 1-aminocyclopropane-1-carboxylic acid (ACC) as the sole nitrogen source. After 24 h incubation at 30 °C, the cells were pelleted and resuspended in 1.5 mL Tris-HCl pH 7.0 buffer. After one wash of the cells with the same buffer, they were lysed by adding 30 µL of toluene. For the enzymatic assay, 200 µL of cell lysate was incubated 15 min at 30 °C with 20 µL ACC 0.5 M. The reaction was stopped by the addition of 1 mL HCl 0.56 M. The mix was centrifuged for 5 min at 20,000× *g*, and 1 mL supernatants were incubated at 30 °C with 800 µL HCl 0.56 M and 300 µL 2,4-dinitrophenylhydrazine. After 30 min of incubation, 2 mL NaOH 2M were added, and the *A*_540_ was measured in a Shimadzu UV-260 spectrophotometer, Kioto, Japan. The α-ketobutyrate produced was determined using a calibration curve built with known concentrations of α-ketobutyrate. The protein concentration of the cell extracts was determined using the Bradford method [54].

### 2.9. Cadmium Concentration in Plants Tissues

Root samples were washed for 15 min with 10 mM Pb(NO_3_)_2_ and immersed for another 15 min in water to exchange surface-bound elements. Leaves were washed twice with Millipore water. Samples were dried at 105 °C for 1 week, weighted again to determine dry weight, and digested for 3 rounds in HNO_3_ suprapur (70–71%) using a heat block (VWR, Leuven, Belgium). For the fourth digestion cycle, the sample was digested in 1 mL HCl suprapur (37%). Finally, the samples were dissolved in 500 µL of 20% HCl, and completed till 5 mL with Millipore water. The metal concentration was determined by inductively coupled plasma atomic emission spectrometry (Perkin-Elmer, 1100B, Waltham, MA, USA). Blanks (only HNO_3_) and a standard sample (NIST Spinach (1570a)) were analyzed for reference purposes.

### 2.10. Superoxide Dismutase (SOD) Enzymatic Assay

SOD activity was estimated according to McCord and Fridovich, 1969. Root samples (50 mg) were collected and immediately frozen in liquid nitrogen and stored at −80 °C. The frozen tissue was homogenized in 0.1 M Tris-HCl buffer pH 7.8, 1 mM EDTA, 1 mM DTT and 4% polyvinylpyrrolidone. After homogenization, the samples were centrifuged for 10 min at 20,000× *g* and 4 °C. The sample solution contained: 50 mM KH_2_PO_4_ pH 7.0 buffer, 1 mM EDTA, 0.1 mM cytochrome *c*, 0.5 mM xanthine and 5 U/mL xanthine oxidase, supplemented with 100 µL homogenized rice roots, or 100 µL of homogenization buffer (standard solution). The inhibition of the increment in the *A*_550_ measured in the sample solution, compared to the standard, was used to estimate the SOD enzyme activity. One unit of enzyme activity was defined as the quantity of SOD necessary to inhibit 50% of the cytochrome *c* reduction [55].

### 2.11. Statistical Analysis

The data were analysed with the program IBM SPSS statistics 24 using analysis of the variance ANOVA. One-way or two-way ANOVA were applied to determine significant differences between the mean obtained in the assays performed. The post hoc Bonferroni test was employed to compare the different conditions. Statistical differences were presented as *p <* 0.001 (***), *p <* 0.01 (**) or *p <* 0.05 (*).

## 3. Results and Discussion

### 3.1. Azoarcus sp. CIB Is Able to Promote Plant Growth

We investigated whether the CIB strain was able to effectively promote the growth of rice plants as endophyte. To do so, rice seeds were germinated and inoculated with *Azoarcus* sp. CIB (pSEVA237) (Table 1), using *E. coli* CC118 (pSEVA237) as a non-endophyte reference bacterium (Table 1). Plants inoculated with *Azoarcus* sp. CIB showed a 30% increment in weight compared to those that were not inoculated (Figure 1). Since the weight of plants inoculated with *E. coli* CC118 did not increase compared to that of non-inoculated plants (Figure 1), we could exclude that the *Azoarcus* sp. CIB-mediated growth promotion was the consequence of a bacterial fertilizer effect. It has been previously reported that *Azoarcus* sp. CIB is able to colonize rice roots as an endophyte, showing several PGP traits such as nitrogen fixation, production of indoleacetic acid, and solubilization of phosphate [30]. However, it is known that only a limited number of bacteria demonstrating in vitro PGP traits are effectively able to promote plant growth [56]. Here, our results confirm that *Azoarcus* sp. CIB can promote the growth of rice plants, hence acting as a PGP-bacterium. In this sense, *Azoarcus* sp. CIB, that is likely a member of the new *Aromatoleum* genus [31], behaves as typical *Azoarcus* strains that colonize plants as endophytes and promote plant growth giving rise to a 50% rice weight increase, e.g., *Azoarcus* sp. DQS-4 [23], or 15% weight increase, e.g., *Azoarcus* sp. BH72 [25,28]. Further studies are in progress to determine the factors and mechanisms involved in the weight increase of rice plants after inoculation with the CIB strain.

### 3.2. Engineering an ACC Deaminase-Producing Azoarcus sp. CIB Strain

To engineer a CIB strain harbouring ACC deaminase activity, the *acdS* gene from *Paraburkholderia phytofirmans* (formerly *Burkholderia phytofirmans* PsJN) was cloned under control of the heterologous *Ptac* promoter generating the promiscuous plasmid pSEVA237acdS (Table 1). The new recombinant *Azoarcus* sp. CIB (pSEVA237acdS) strain did not show any significant difference in growth rate in the culture medium compared with that of the wild type CIB strain (data not shown), suggesting that expression of *acdS* did not reduce bacterial fitness. However, whereas the wild-type CIB strain did not exhibit detectable ACC deaminase activity, the recombinant *Azoarcus* sp. CIB (pSEVA237acdS) strain displayed a significant deaminase activity (6716 nmol mg^−1^ h^−1^), almost two times higher than that observed in the parental *P. phytofirmans* PsJN strain (3716 nmol mg^−1^ h^−1^) (Figure 2). This high ACC deaminase activity in *Azoarcus* sp. CIB (pSEVA237acdS) could be due to the fact that the *acdS* gene is in multicopy doses and expressed under the strong *Ptac* promoter. The ACC deaminase activity level is in the range of that previously reported for *Rhodococcus* sp. strain 4N-4 (12,970 nmol mg^−1^ h^−1^), *Rhodococcus* sp. strain Hp2 (7320 nmol mg^−1^ h^−1^), *Variovorax paradoxus* 5C-2 (4322 nmol mg^−1^ h^−1^) or *Acidovorax facilis* 4p-6 (3080 nmol mg^−1^ h^−1^) strains [57], but much higher than that reported for most microorganisms equipped with ACC deaminase enzyme [6].

Endophytic bacteria that promote plant growth often harbour the *acdS* gene [58], which codes for an ACC deaminase that can lead to lower levels of the plant growth-inhibiting ethylene [11]. Interestingly, none of the *Azoarcus* strains described so far, including *Azoarcus* sp. CIB, possess the *acdS* gene [13,59]. Therefore, it could be interesting to examine the performance of an Azoarcus/Aromatoleum strain expressing an exogenous *acdS* gene. The heterologous expression of ACC deaminase genes and their biological effects have been reported previously in some bacteria. For example, the impact of ACC deaminase overproduction on plant growth and enhancement of nodulation abilities by *Cupriavidus taiwanensis* STM894 [60], *Serratia grimesii* BXF1 [61] and *Mesorhizobium ciceri* strain LMS-1 [19,62] has been described. Also, it has been reported the positive effects on plant growth promotion and plant copper tolerance of a genetically engineered *Sinorhizobium meliloti* strain overproducing ACC deaminase [20] have been reported, as well as the enhancement of the symbiotic performance of *Mesorhizobium ciceri* strains overproducing ACC deaminase when they were used as inoculants of chickpea plants cultivated under saline conditions, thus alleviating the negative effects caused by salinity [21]. However, to the best of our knowledge, our results evidence the first example of successful expression of an *acdS* gene in an endophyte bacterium of the Rhodocyclaceae family. Moreover, the broad-host-range plasmid pSEVA237acdS becomes a new and interesting genetic tool to transfer the ACC deaminase activity to other plant-associated bacteria that do not possess this activity.

### 3.3. Azoarcus sp. CIB-Expressing acdS Gene Promotes Rice Growth under Stress Conditions

To check whether the expression of the *acdS* gene in *Azoarcus* sp. CIB confers an additional PGP trait to this recombinant bacterium, we took advantage of the ability of strain CIB to tolerate moderate concentrations of the toxic metal Cd (up to 1 mM CdCl_2_) without significant effects on cell growth [36]. As expected, non-inoculated rice seedlings exposed to CdCl_2_ (50 µM) showed a significant weight reduction (Figure 3A). However, the weight of Cd-exposed plants inoculated with *Azoarcus* sp. CIB (pSEVA237acdS) (0.23 ± 0.03 g) was 65 ± 3% higher compared to that of non-inoculated control plants or plants inoculated with the CIB (pSEVA237) control strain (0.15 ± 0.02 g) (Figure 3B). Thus, a protective effect was obvious in case of Cd-exposed plants inoculated with *Azoarcus* sp. CIB (pSEVA237acdS) expressing the ACC deaminase gene. Hence, these results demonstrated that the expression of the *acdS* gene in *Azoarcus* sp. CIB (pSEVA237acdS) colonizing rice roots may protect the plant against the toxic effects of Cd exposure.

Endophytic bacteria expressing ACC deaminase activity have been associated with the protection of the plants growing in hostile environments that provoke ethylene-induced stress [6], and the relationship between metals and ethylene has been well established in transgenic plants expressing a bacterial *acdS* gene [2,63]. Moreover, it has been described previously that some bacteria with ACC deaminase activity can promote plant growth under metal stress [64] or drought stress [65]. For example, the protective effects of *Burkholderia* sp. J62 on Pb (lead)- and Cd -exposed *Lycopersicon esculentum* [66], *Pseudomonas putida* UW4 on Ni (nickel)-exposed *Brassica napus* [67], *Pseudomonas putida* UW4 on Cu (copper)-exposed *Medicago lupulina* [20], *Variovorax paradoxus* on Cd-exposed *Brassica juncera* [57] have been reported. Concerning rice, it has been shown that plants inoculated with mutant bacteria lacking the *acdS* gene and grown in soil polluted with zinc were not able to develop normally, in contrast to plants inoculated with wild-type bacteria expressing the *acdS* gene [68]. As mentioned above, Cd is a very harmful metal, even at low concentrations, that can accumulate in soils leading to phytotoxic effects and hence decrease agricultural production [69]. It is well-known that Cd provokes stress in plants by increasing levels of reactive oxygen species (ROS) and ethylene production [9,70,71]. In fact, under laboratory conditions, rice seedlings were unable to grow when exposed to 500 µM CdCl_2_, and a significant (about 50%) reduction of growth was observed at 50 µM (Figure 3A). Therefore, to the best of our knowledge, this is the first report about PGPB-expressing ACC deaminase protecting rice plants to stress induced by Cd exposure.

### 3.4. Enhanced Cd Concentration in the Shoots by Strain CIB Expressing the acdS Gene

To determine the effects of *Azoarcus* sp. CIB inoculation on Cd concentrations and its allocation in rice, the total metal concentrations in roots and shoots were determined. No significant differences in root Cd concentrations were observed when comparing treatments with or without bacterial inoculations (Figure 4A).

However, the shoots of plants inoculated with *Azoarcus* sp. CIB (pSEVA237acdS) showed significantly higher Cd concentrations (38.24 ± 6) than the non-inoculated Cd-exposed controls (19.3 ± 1.4) (Figure 4B). Interestingly, the *Azoarcus* sp. CIB (pSEVA237) strain, which does not express the *acdS* gene, also increased the metal concentration in the shoots compared to the non-inoculated control, leading to Cd concentrations that were not significantly different from those obtained with the recombinant CIB (pSEVA237acdS) strain. Hence, independent of the *acdS* gene expression, *Azoarcus* CIB can alter metal concentrations in tissues, which is an interesting finding for phytoextraction applications. Although the leaf Cd concentrations in the range of 15–40 µg g^−1^ are toxic to rice [72], as seen by the plant growth inhibition (Figure 3), *Azoarcus* sp. CIB (pSEVA237acdS) inoculated plants were growing better (Figure 3), which suggests that for the same concentration of metals in the leaves, the plants inoculated with CIB expressing *acdS* can cope better with the metals.

It has been described that rhizosphere residing bacteria equipped with ACC deaminase activity are able to enhance the uptake of inorganic pollutants through modification of the root architecture and the root uptake system of the plant, increasing the accumulation of the metal in plant tissues [57,73]. In addition, other mechanisms by which PGPB can influence Cd-uptake have been extensively reviewed [74] and include for example, microbe induced changes to the rhizosphere pH, Cd solubilisation, organic acid secretion, by which Cd can be more readily available in the rhizosphere for passive or active uptake by root cells [74,75]. Similar mechanisms could explain the effect of the endophytic *Azoarcus* sp. CIB strain expressing the exogenous *acdS* gene using rice as proof of concept of phytoextraction and survival at high concentrations of Cd. Although rice is not commonly recommended for cultivation on Cd-polluted paddy soils since Cd can be readily incorporated by plants [76], and easily found in rice grains [77] and incorporated in the food chain [78], some reports showed that specific varieties of rice, such as TCM213, can be possible candidates for effective Cd phytoextraction [41]. The use of rice has been especially suggested for paddy soils, environments that are more adapted for rice cultivation [42,43]. In this sense, it has also been also described that in soil polluted with zinc, some endophytes were able to increase the rice growth and Zn concentration in rice tissues [69]. Finally, it is worth noting that plants growing in oil hydrocarbons polluted soils are often subjected to strong stress linked to the production of an excess of ethylene [79]. The ability of *Azoarcus* sp. CIB to degrade aromatic compounds, such as toluene and *m*-xylene, paves the way to employ the CIB strain expressing the *acdS* gene as a strategy to promote rice plant growth both by eliminating the aromatic pollutants from the paddy soils by controlling the ethylene levels in the plant.

### 3.5. Azoarcus sp. CIB Expressing acdS Gene May Increase the ROS Quenching Capacity in Rice Roots

To further explore the mechanisms behind the plant growth promotion by *Azoarcus* sp. CIB (pSEVA237acdS) cells, we estimated the levels of oxidative stress in the plants using the activity of superoxide dismutase (SOD) as an indicator of superoxide levels in roots. As expected, exposure of non-inoculated plants to 50 µM CdCl_2_ caused an important increase of the SOD activity (66 ± 6%) (Figure 5). However, when plants were inoculated with strain CIB (pSEVA237), SOD activity was lower, being the lowest when the strain used for inoculation was equipped with the *acdS* gene, i.e., *Azoarcus* sp. CIB (pSEVA237acdS) (Figure 5). It is well known that plant responses to metal-exposure are directly linked to the ROS produced and the levels of ethylene [9].

The results presented here show that ACC deaminase activity is mitigating the SOD activity, which suggests a reduction of ROS levels in CIB (pSEVA237acdS) inoculated rice. These results are in agreement with previous work reporting decreases of the SOD activity in plants exposed to different concentrations of metals and aromatic compounds when inoculated with PGPB-expressing ACC deaminase genes [80,81]. Therefore, the growth-promoting effect observed with strain *Azoarcus* sp. CIB (pSEVA237acdS) under Cd exposure can, at least partly, be attributed to its modulating effects on the levels of ethylene produced by the plant and the reduction of ROS levels.

## 4. Conclusions

The results of the present study allow us to conclude that an aromatics-degrading and metal-resistant strain closely related to endophytic Azoarcus, i.e., *Azoarcus* sp. CIB, is able to produce a 30% plant weight increase when colonizing rice as an endophyte, hence revealing that it behaves also as a PGP-bacterium. In addition, the heterologous expression of an exogenous ACC deaminase gene in the CIB strain generates a PGP that increases the weight of rice plants growing in a Cd-polluted semi-solid medium, indicating its ability to protect rice against the Cd-induced stress, and increasing the concentration of Cd in rice tissues. To the best of our knowledge, this is the first report about the use of PGPB-expressing ACC deaminase to protect rice plants to the stress induced by Cd exposure. The pSEVA237acdS plasmid developed in this work represents a new genetic tool that could be implemented to provide an additional PGP trait in those PGPB that lack ACC deaminase activity, hence fostering plant production even under stress conditions such as cultivation in polluted soils. In addition, we open here a new possibility to explore the use of rice for phytoremediation of Cd-polluted paddy soils. Since Cd-polluted soils often also contain oil hydrocarbons, the application of *Azoarcus* sp. CIB (pSEVA237acdS) as a tool to improve the growth of rice, the phytoextraction of Cd and the removal of aromatic pollutants will be investigated.

## Figures and Tables

**Figure 1 microorganisms-08-01453-f001:**
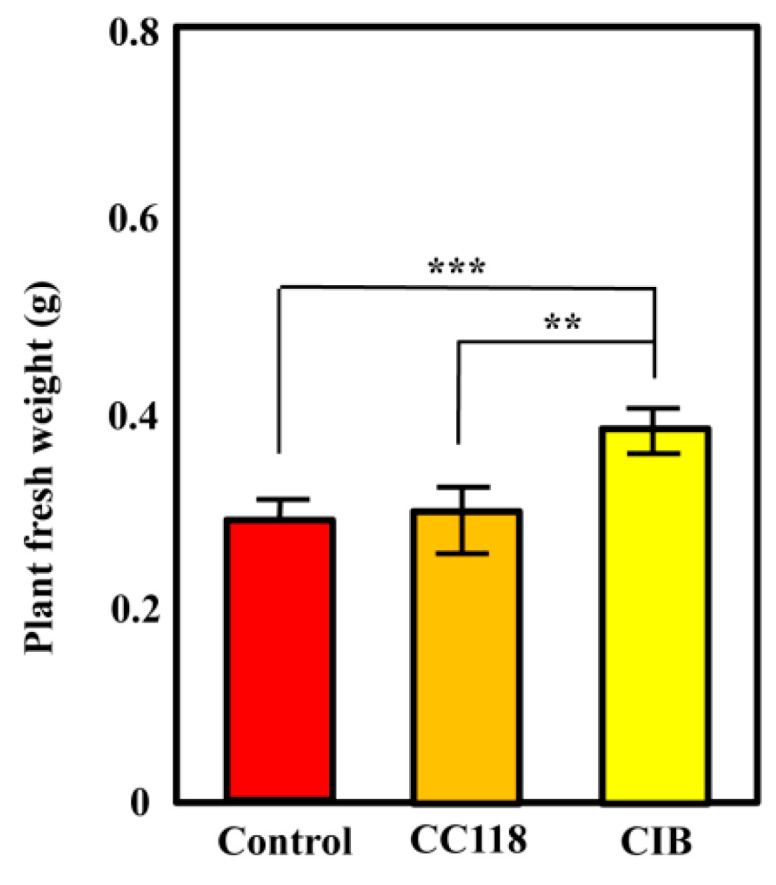
Ability of *Azoarcus* sp. CIB to promote the growth of rice. Rice seedlings inoculated with *E. coli* CC118 (pSEVA237) (CC118) (orange column), *Azoarcus* sp. CIB (pSEVA237) (CIB) yellow column), or non-inoculated seedlings (Control) (red column), were grown for 4 weeks in a growth chamber under controlled conditions. Total fresh weight of each plant in grams (g) was determined. Mean values (20 plants per each condition) are presented with the corresponding errors. The asterisks indicate the statistical significance observed when two conditions are compared applying the One-way ANOVA method and the Bonferroni post hoc test. *p <* 0.001 (***); *p <* 0.01 (**).

**Figure 2 microorganisms-08-01453-f002:**
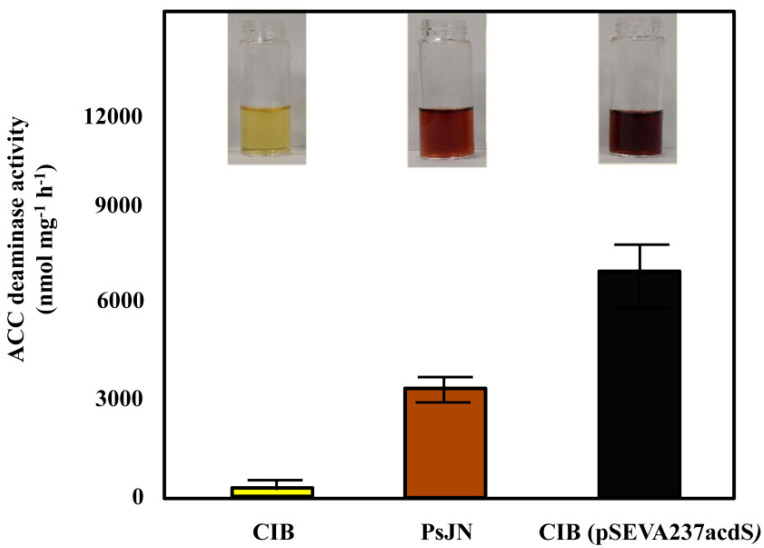
ACC deaminase activity in different bacterial strains. The ACC deaminase activity of *Azoarcus* sp. CIB (yellow column), *P. phytofirmans* PsJN (brown column) and *Azoarcus* sp. CIB (pSEVA237acdS) (dark grey column) was determined according to a colorimetric method described in Section 2. The intensity of the color varies from the yellow (negative) to dark brown (positive). The spectrophotometric method determines the presence of α-ketobutyrate (nmol per mg of total protein^−1^ h^−1^) produced from ACC. Means of three independent experiments are presented and the standard deviation is indicated as error bars.

**Figure 3 microorganisms-08-01453-f003:**
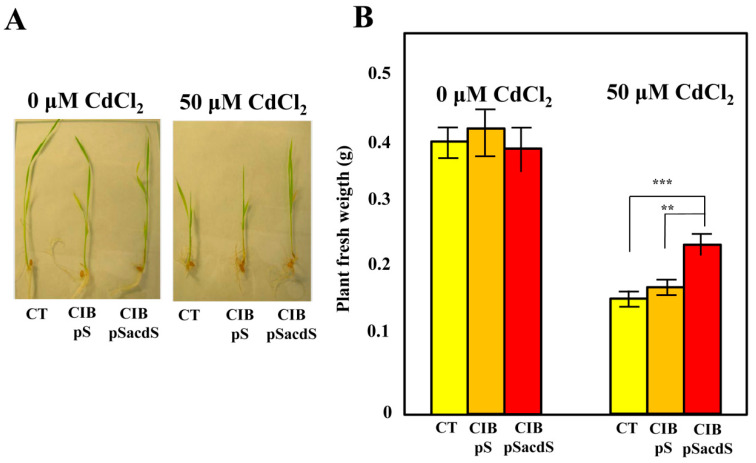
Analysis of the ability of *Azoarcus* sp. CIB producing ACC deaminase to protect rice against the stress induced by Cd exposure. (**A**) Seedlings of rice obtained after inoculation with *Azoarcus* sp. CIB (pSEVA237) (CIB pS) (red column), *Azoarcus* sp. CIB (pSEVA237acdS) (CIB pSacdS) (orange column), or without bacteria (CT) (yellow column), after 10 days of growth in substrates containing 0 µM or 50 µM CdCl_2_ in greenhouse conditions. (**B**) Rice plants inoculated with *Azoarcus* sp. CIB (pSEVA237) (CIB pS) (red column), *Azoarcus* sp. CIB (pSEVA237acdS) (CIB pSacdS) (orange column), or without bacterial inoculation (CT) (yellow column), were grown in substrates containing 0 µM or 50 µM CdCl_2_ in greenhouse conditions as detailed in Section 2. The total fresh weight of each plant (g) was determined. Presented are mean values for 20 plants with the corresponding errors. The asterisks indicate the statistical significance observed when two conditions were compared applying the Two-ways ANOVA method and the Bonferroni test. *p <* 0.001 (***); *p <* 0.01 (**).

**Figure 4 microorganisms-08-01453-f004:**
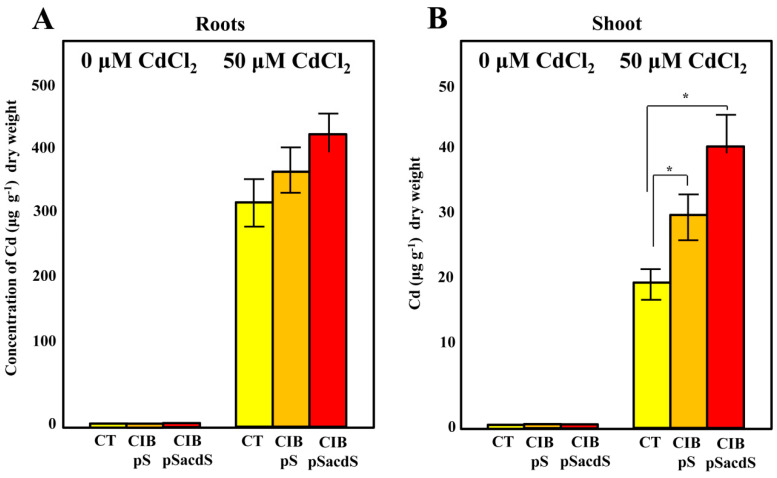
Cd concentrations in roots (**A**) and shoots (**B**) of rice. Rice plants inoculated with *Azoarcus* sp. CIB (pSEVA237) (CIB pS) (orange column), *Azoarcus* sp. CIB (pSEVA237acdS) (CIB pSacdS) (red column), or without bacterial inoculation (CT) (yellow column), were grown in substrates containing 0 µM or 50 µM CdCl_2_ in greenhouse conditions as detailed in Section 2. Total Cd concentrations (µg/g dry weight) of the harvested plants were determined by ICP-AES. Presented are mean values for 5 plants with the corresponding errors. The asterisks indicate the statistical significance observed when the three conditions were compared applying the one-way ANOVA method and the Bonferroni test. *p <* 0.05 (*).

**Figure 5 microorganisms-08-01453-f005:**
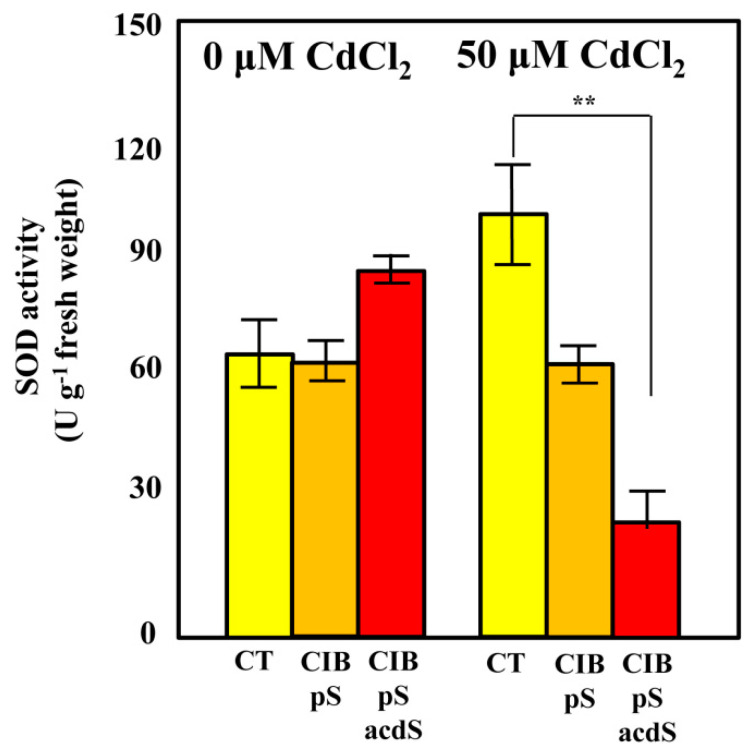
Superoxide dismutase activity in rice roots. Rice plants were grown in Hoagland’s medium in the absence (0 µM) or presence (50 µM) of CdCl_2_ after inoculation with *Azoarcus* sp. CIB (pSEVA237) (CIB pS) (orange columns), *Azoarcus* sp. CIB (pSEVA237acdS) (CIB pSacdS) (red columns), or without bacterial inoculation (CT) (yellow column) in greenhouse conditions as detailed in Section 2. After harvesting, the superoxide dismutase activities were estimated in roots. Data show the mean of 10 independent measurements; the standard deviations are presented as error bars. Asterisks show the statistical significance after applying the two-way ANOVA analysis and the Bonferroni test. *p <* 0.01 (**).

**Table 1 microorganisms-08-01453-t001:** Bacterial strains and plasmids used in this study.

Strain or Plasmid	Relevant Genotype and Characteristic(s)	Reference or Source
***E. coli* strains**		
*DH10B*	F’, *mcrA, Δ(mrr hsdRMS-mcrBC*), *Φ80lacZΔM15, ΔlacX74,* *deoR, recA1*, *araD139,* *Δ**(ara-leu)7697, galU, galK*, *rpsL* (Sm^r^), *endA1, nupG*	Life Technologies
S17-1λ*pir*	Tp^r^ Sm^r^ *recA thi hsdRM^+^* RP4::2-Tc::Mu::Km λ*pir* phage lysogen	[44]
CC118	*Δ**(ara-leu), araD, Δ**lacX7, galE, galK, phoA20, rpoB thi-1, rpsE*, (Sp^r^), (Rf^r^), *argE*, (Am), *recA1*	[45]
***Azoarcus* strains**		
CIB	Wild type strain	[32]
***Paraburkholderia* strains**		
*P. phytofirmans PsJN*	Wild type strain	[46]
***Plasmids***		
pSEVA237	Km^r^, ori pBBR1, harbors the gfp gene under the control of the PlexA promoter	[47]
pSEVA237acdS	Km^r^, pSEVA237 derivative that includes a AscI fragment of 1172 bp containing the gene *acdS* under the control of *Ptac* promoter	This work
pIZ1016	Gm^r^, *ori* pBBR1MCS-5 derivative vector for cloning and expression harboring the *Ptac* promoter and the *lacI* gene	[48]
pIZacdS	Gm^r^, pIZ1016 derivative containing the gene *acdS* from *P. phytofirmans under the control of the Ptac promoter*	This work

Km^r^: kanamycin resistant; Gm^r^: gentamicin resistant; Sm^r^: streptomycin resistant.

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
