# Peer review of "Enhancing the Rice Seedlings Growth Promotion Abilities of Azoarcus sp. CIB by Heterologous Expression of ACC Deaminase to Improve Performance of Plants Exposed to Cadmium Stress"

_microorganisms, 2020, doi:10.3390/microorganisms8091453_

Round 1
Reviewer 1 Report
The manuscript deals with an important topic, tolerance and alleviation of heavy metals stresses by inoculation of Plant Growth Promoting Bacteria.
The topic is of great interest in the context of soil preservation and corresponds to Microorganisms expectations
The methods carried out are sound and currently used. The manuscript is well written, well illustrated (figure are of great interest) and structurated.
Some remarks, nevertheless, should be considered
Currently, under stressed conditions, fresh and dry weight of aerial and roots parts are measured and the ratio of dry weight (root/aerial) is calculated in order to show how plants adapte their strategie to grow. This would be interesting to depict the adaptative strategies developed by each plant (control and inoculated).
Moreover, authors can refer to recent studies to use PGPB for other uses.
Danish, S.; Zafar-Ul-Hye, M.; Hussain, S.; Riaz, M.; Qayyum, M.F. Mitigation of drought stress in maize through inoculation with drought tolerant ACC deaminase containing PGPR under axenic conditions. Pakistan J. Bot. 2020, 52, 49–60.
Danish, S.; Zafar-ul-Hye, M. Combined role of ACC deaminase producing bacteria and biochar on cereals productivity under drought. Phyton (B. Aires). 2020, 89, 217–227.
Danish, S.; Zafar-ul-Hye, M.; Mohsin, F.; Hussain, M. ACC-deaminase producing plant growth promoting rhizobacteria and biochar mitigate adverse effects of drought stress on maize growth. PLoS One 2020, 15, e0230615.
Author Response
Q1: Currently, under stressed conditions, fresh and dry weight of aerial and roots parts are measured and the ratio of dry weight (root/aerial) is calculated in order to show how plants adapte their strategie to grow. This would be interesting to depict the adaptative strategies developed by each plant (control and inoculated).
A1: Thank you for the comment. The presentation of the plant weight data in the manner than the reviewer suggests can be very informative and we will consider it for future works. Nevertheless, since in our previous reports describing that CIB is an endophytic bacterium we used total plant weight measurements (Fernández-Llamosas, H. et al. (2014) PLoS One 9, e110771), we decided also to keep these type of measurements in this work to facilitate comparisons.
Q2: Moreover, authors can refer to recent studies to use PGPB for other uses.
Danish, S.; Zafar-Ul-Hye, M.; Hussain, S.; Riaz, M.; Qayyum, M.F. Mitigation of drought stress in maize through inoculation with drought tolerant ACC deaminase containing PGPR under axenic conditions. Pakistan J. Bot. 2020, 52, 49–60.
Danish, S.; Zafar-ul-Hye, M. Combined role of ACC deaminase producing bacteria and biochar on cereals productivity under drought. Phyton (B. Aires). 2020, 89, 217–227.
Danish, S.; Zafar-ul-Hye, M.; Mohsin, F.; Hussain, M. ACC-deaminase producing plant growth promoting rhizobacteria and biochar mitigate adverse effects of drought stress on maize growth. PLoS One 2020, 15, e0230615.
A2: Thank you for the comment. The reviewer is right, it would very informative to mention other conditions were ACC deaminase produce plant growth promotion such mitigating the negative effect of the stress promoted by the drought. We have included one reference indicating that point (line 347 and reference #65).
Reviewer 2 Report
It is a publication of high scientific quality and importance. The only thing I would like to point out is that it is too little to assess the intensity of oxidative stress by SOD alone. Occasionally, higher SOD activity coincides with decreased superoxide concentration. More over, activities of other antioxidant enzymes and concentrations of ROS and antioxidative metabolites are important. They are closely related, their relationship is strongly determined by species, growth stage, stressor and its intensity, location in the plant and even in the cell, etc. Thus, the effect of PGPB on the intensity of oxidative stress in plants based on a single antioxidant enzyme should be assessed with caution.Author Response
Q: It is a publication of high scientific quality and importance. The only thing I would like to point out is that it is too little to assess the intensity of oxidative stress by SOD alone. Occasionally, higher SOD activity coincides with decreased superoxide concentration. Moreover, activities of other antioxidant enzymes and concentrations of ROS and antioxidative metabolites are important. They are closely related, their relationship is strongly determined by species, growth stage, stressor and its intensity, location in the plant and even in the cell, etc. Thus, the effect of PGPB on the intensity of oxidative stress in plants based on a single antioxidant enzyme should be assessed with caution.
A: Regarding the ROS/SOD: the reviewer is right, the measure of SOD is an indirect method to measure ROS. That is the reason why have toned down this assessment by indicating at the title of paragraph: “Azoarcus sp. CIB expressing acdS gene may increase the ROS quenching capacity in rice roots”. For that reason we are planning to deeply analyze other enzymes and conditions related with ROS in the future.